# Oligomeric Formulas in Surgery: A Delphi and Consensus Study

**DOI:** 10.3390/nu13061922

**Published:** 2021-06-03

**Authors:** Daniel De Luis Román, Eduardo Domínguez Medina, Begoña Molina Baena, Pilar Matía-Martín

**Affiliations:** 1Center of Investigation Endocrinology and Nutrition, Hospital Clínico Universitario de Valladolid, Medicine School University, 47002 Valladolid, Spain; dluisro@saludcastillayleon.es; 2Center for Research in Molecular Medicine and Chronic Diseases (CiMUS), 15782 Santiago de Compostela, Spain; eduardo.dominguez@usc.es; 3Hospital Universitario de la Princesa, 28006 Madrid, Spain; bmolinabaena@gmail.com; 4Endocrinology and Nutrition Unit, Hospital Clínico San Carlos, Instituto de Investigación Sanitaria del Hospital Clínico San Carlos (IdISSC), 28040 Madrid, Spain

**Keywords:** oligomeric formula, enteral nutrition, surgery, short bowel syndrome, malabsorption

## Abstract

Nutritional management of patients with intestinal failure often includes the use of oligomeric formulas. Implementing the use of oligomeric formulas in surgical patients with maldigestion or malabsorption could be a nutritional strategy to be included in clinical protocols. We aim to generate knowledge from a survey focused on the effectiveness of nutritional therapy with oligomeric formulas with Delphi methodology. Each statement that reached an agreement consensus among participants was defined as a median consensus score ≥7 and as an interquartile range ≤3. The use of oligomeric formulas in surgical patients, starting enteral nutrition in the post-operative phase in short bowel syndrome and in nonspecific diarrhea after surgical procedures, could improve nutritional therapy implementation. Stakeholders agreed that early jejunal enteral nutrition with oligomeric formula is more effective compared to intravenous fluid therapy and it is useful in patients undergoing upper gastro-intestinal tract major surgery when malabsorption or maldigestion is suspected. Finally, oligomeric formulas may be useful when a feeding tube is placed distally to the duodenum. This study shows a practical approach to the use of oligomeric formulas in surgical patients with intestinal disorders and malabsorption, and it helps clinicians in the decision-making process.

## 1. Introduction

Evidence obtained from observational data generated during routine clinical practice or Real-World Data (RWD) is an important source for supporting clinical decisions about nutritional interventions. Although physicians could take advantage of RWD for their informed decisions, data are not always accessible or clinically relevant [1]. 

Intestinal failure (IF)—the reduction of gut function below the minimum necessary for the absorption of macronutrients and/or water and electrolytes, such that intravenous supplementation is required to maintain health and/or growth—and intestinal insufficiency (II)—the reduction of gut absorptive function that does not require intravenous supplementation to maintain health and/or growth [2]—are severe medical conditions intimately related, among others, to short bowel syndrome (SBS), which is a malabsorption disorder caused by a lack of functional small intestine, resulting in an inadequate fluid and nutrient absorption [3]. 

The origin of malnutrition in intestinal diseases is multifactorial and includes a deficient intake of nutrients, maldigestion and malabsorption due to different clinical conditions, with surgery being one of the most involved etiologies in these clinical situations. These alterations usually cause abdominal discomfort, diarrhea, weight loss and malnutrition. Thus, prevention and correction of nutritional deficit with the assessment of nutritional status is critical and should be part of the multidisciplinary management of these patients [4].

Polymeric enteral nutrition (EN) is the most used formula for enteral nutritional support; however, EN with oligomeric formulas containing peptides and medium chain triglycerides (MCT) is part of the management of patients with intestinal diseases and IF or II, including SBS [5,6], and can facilitate the absorption of nutrients in case of impaired intestinal function or anatomical change after surgery [7,8]. The use and implementation of clinical protocols for nutritional treatment of IF or II may facilitate the standardization of care and decision-making with regards to the use of oligomeric formulas in these patients.

The European Society for Clinical Nutrition and Metabolism (ESPEN) has established “special interest groups” devoted to developing guidelines for the formal definition and classification of IF [9,10]. These recommendations are relevant to approach decision-making situations in the clinical management of these patients.

The combination of data information and clinical experience is the standard procedure to demonstrate the effectiveness of specific interventions, such as the use of oligomeric formulas, nutrition monitoring and surveillance and the assessment of metabolic status.

The aim of this study is to support an evidence-based decision-making process of oligomeric formulas in patients with intestinal disorders after surgery in the context of a surgeon’s experience. 

## 2. Materials and Methods

This study was carried out by using the Delphi method. In Spain, this study does not require Research Ethics Committees (RECs) approval or written consent from patients.

We elaborated a prospective consensus study using a Delphi process. Delphi is a structured process effective to evaluate the cost, effectiveness, applicability and sustainability in a medical setting and it is widely used in developing core outcomes sets. The Delphi method aims to achieve consensus through the collection of stakeholder opinions and is especially useful in those clinical situations in which there is great uncertainty due to a lack of information or consensus [11].

### 2.1. Phase 1

A coordinating group was set up to inform the development of the various stages of this study and to discuss the results at each phase. The coordinating group, formed by endocrinologists with a strong background in clinical nutrition, raised, defined and justified existing controversies about the use of oligomeric formulas in intestinal diseases. To develop a preliminary list of assessments for a Delphi survey, stakeholders performed a literature review about the prescription of oligomeric diets in patients with intestinal diseases and existing controversies in clinical practice.

### 2.2. Phase 2

The coordinating group prepared a document with 13 clinical questions focused on the use of oligomeric formulas in patients that had undergone digestive surgery for intestinal diseases, including clinical conditions such as hyper-secretion phase in SBS, postoperative upper digestive tract, nonspecific diarrhea, gastrectomy, pancreaticoduodenectomy, fistulas and colon resection.

### 2.3. Phase 3

The study was carried out with the participation of 47 surgeons with extensive experience in digestive surgery and nutrition, from 35 different Spanish hospitals (see Acknowledgments). Each of the participants engaged voluntarily in the study and their answers were coded to keep them anonymous and confidential. 

A document explaining the project and survey was sent out to participants, and the Delphi survey consisted of two consecutive rounds. In round 1, a document explaining the purpose of the study, the Delphi method and how to complete the survey was sent to the panelists. All participants were asked to score each assessment and to identify any additional important assessment that did not appear in the list. To evaluate the pertinence to the survey, the coordinating group reviewed additional assessments added in round 1.

The first round remained open for 6 weeks with a 4-week reminder sent to those surgeons that had partially completed or not completed the questionnaire. At the end of the first round, after 6 weeks, all participants completed the survey and their answers were coded and anonymized in a database.

### 2.4. Phase 4

In round 2, the results from round 1 were analyzed. Following the analysis, the results were reviewed by the coordinating group and considered comparable in terms of percentage spread across the responses of 1–9. Assessments without consensus were revised and rephrase in a dichotomic way (Yes/No).

### 2.5. Scoring Method

In each round, panelists were asked to indicate their level of agreement with each statement using a nine-point scale (1: “strongly disagree” to 9: “strongly agree”). Scores 1–3 (rank 1) were considered as having a low degree of agreement, implementation, or knowledge according to the question, scores 4–6 (rank 2) were considered as doubtful and scores 7–9 (rank 3) were considered as having a high degree of agreement.

### 2.6. Methods of Analysis

For each round, descriptive statistics were used to summarize the results for each question, including the percentage of participants scoring from 1 to 9. Each statement that achieved an agreement consensus was defined as a median consensus score (MED) of ≥7, and as an interquartile range (IQR) of ≤3. Likewise, a MED score of ≤3 was considered as a consensus to refuse the statement, while an IQR of ≥4 or a MED of 4–6 was considered as no agreement. Those statements were reviewed and included in the second round with only two possible answers (Yes/No) and we considered general consensus when 50% or more of the answer were “Yes” or “No”.

### 2.7. Data Analysis

Statistical analysis was performed with the statistical software IBM-SPSS version 23 (Chicago, IL, USA). MED, IQR and Standard Deviation were calculated for each statement. A comparison of variables was performed with a non-parametric U-Mann Whitney test and a *p*-value < 0.05 was regarded as significant.

## 3. Results

### 3.1. Participant Characteristics

A total of 47 panelists completed the survey (76.6% men). Most worked in a General and Digestive Surgery Unit (66%) and had an assistant position (63.8%). Furthermore, most surgeons (37%) worked in medium-large hospitals with 500–1000 hospital beds (Table 1).

### 3.2. Descriptive Statistics

The descriptive statistics of the 13 statements and the following consensus among the experts are shown in Table 2.

Globally, the survey showed a lack of knowledge about ESPEN recommendations [9] by most of the surgeons (MED 3, with 53.2% of answers in rank 1 score 1–3). The degree of application of this article among the surgeons surveyed was also low (MED 3, with cumulative percentage in rank 1 in 55.3% of them). 

We found a high application of ESPEN clinical guidelines with regards to the use of a small number of oligomeric formulas as an option to start enteral or oral nutrition in the post-operative hyper-secretion phase in SBS in the case of gastrointestinal intolerance of polymeric formulas (with 80.9% of the surgeons consulted (n = 38) scoring 7 or higher—MED 8).

Most of the experts (68.1%) agreed that oligomeric low-fat formulas could be used in patients with nonspecific diarrhea (MED 7), while only 6.4% disagreed. 

Surgeons reached a positive consensus (MED 7) with regard to the use of oligomeric formulas in patients undergoing gastrectomy, especially with a Roux-en-Y anastomosis and maldigestion. We found that 72.4% (n = 34) of surgeons awarded a score of 7 or more, while only 4.3% (n = 2) disagreed.

A positive consensus was achieved regarding the usefulness of oligomeric formulas in patients treated with cephalic pancreaticoduodenectomy with maldigestion, with 76.6% (n = 36) of experts awarding a score of 7 or more (MED 8), and also in patients with low-debit ileus or colon fistulas, with 89.4% of surgeons (n = 52) awarding a score of 7 or higher (MED 8). Furthermore, most experts (72.3%, n = 34) agreed (MED 8) that oligomeric formulas are well tolerated in patients treated with right hemicolectomy or total colectomy.

Surgeons surveyed did not reach a consensus in the first round about the statement “ESPEN guidelines recommend the use of needle catheter jejunostomy (NCJ) in patients undergoing major upper gastrointestinal and pancreatic surgery,” with 19 physicians (40.5%) giving a score of 3 or less and 19 (40.5%) giving a score of 7 or higher. In order to reach a consensus, the same question was asked again in a dichotomous way (yes/no), and a consensus of no agreement was reached by simple majority (53.2%, n = 25 versus 46.8%, n = 22) in the second round. 

Likewise, participants did not reach a consensus in the first round regarding the statement “In clinical practice, it is common to place a feeding tube distally to the anastomosis in the surgical act,” with 36.2% (n = 17) giving a score of 3 or less and 31.9% (n = 15) giving a score of 7 or higher. In the second round, the same question was asked again but with only two possible answers (yes/no). A consensus of “no agreement” was reached by simple majority (55.3%, n = 26 versus 44.7%, n = 21).

In total, 80.9% of the experts (n = 38) agreed (MED 8) that after upper digestive tract surgery, oligomeric formula may be useful when EN is carried out distally to the duodenum and malabsorption or malnutrition are suspected. 

We also found a positive consensus concerning early jejunal EN with oligomeric formulas providing more nutrients with less weight loss compared to intravenous fluid therapy (85.1%, n = 40 scored 7 or higher with MED 9) and is useful in patients undergoing upper gastro-intestinal tract major surgery (82.9%, n = 39, agreed with MED 8).

### 3.3. Stratified Analysis

#### 3.3.1. Analysis According to the Existence of a Specialized Nutrition Unit

We correspondingly perform a descriptive analysis according to whether the hospital had a specialized Nutrition Unit or not (Table 3), and in all the explored items, no significant differences were detected among the answers (*p* = 0.113).

#### 3.3.2. Analysis According to the Existence of a Clinical Protocol to Approach the Diagnosis and Intervention of Patients with SBS

Table 4 shows the descriptive statistics according to the existence of a clinical protocol to approach diagnosis and intervention of patients with SBS. Stratifying the hospitals, according to the existence (n = 13) or not (n = 33) of a clinical protocol, a significant difference was observed between centers with a protocol for SBS and those that did not only for question 2 (*p* < 0.001). In those centers with clinical protocols, we observed a high degree of agreement (MED 7), with 77% of experts scoring 7 or higher. 

## 4. Discussion

Conclusive results to support the implementation of clinical protocols is often difficult to achieve in real-world settings. The purpose of this study was to reach a consensus that would lay the basis for the use of oligomeric formula in the management of this population of surgical patients using the Delphi method. We surveyed the knowledge on relevant consensus recommendations as well as the clinical use of oligomeric formulas in intestinal diseases associated with IF, II or intestinal surgery.

Polymeric enteral formula contains whole proteins, needing gastro-intestinal tract digestion while oligomeric formula contains hydrolyzed proteins, known as oligopeptides. These peptides have a different uptake transport mechanism that allows an improvement of intestinal absorption compared to whole proteins. Therefore, the use of oligomeric formulas in patients with intestinal surgery should be considered in clinical protocols.

We noticed an unexpected lack of knowledge of ESPEN recommendations [9] by most surgeons surveyed. This ESPEN article aims to give a formal definition and recommendations for the classification of both acute and chronic IF to simplify the collaboration and multidisciplinary management between professionals in clinical practice. It is also important to note that only 13 centers had clinical protocols for SBS, although they were applied at a high degree.

In this survey, surgeons commonly accepted the use of oligomeric EN when polymeric formulas are not tolerated in the postoperative phase of SBS, according to published guidelines [6]. Another narrative review [3] stated that high-output stomas may benefit when elemental diets are chosen, but doubt exists regarding the effect that polymeric or oligomeric formula have on intestinal adaptation after surgery. In animal models, the signs of mucosal regeneration (DNA content in the cells, length of villi and crypt depth) were more evident in polymeric formula fed animals [15], but in patients with high output jejunostomy, an improvement in the absorption of nitrogen was observed after oligomeric diet, without differences in energy or fat absorption [16]. The evidence is poor and, although more trials are needed, the experienced surgeons agree with this statement in clinical practice. 

A positive consensus regarding the use of oligomeric formula was reached in nonspecific diarrhea. It has been stated that in persistent diarrhea, suggesting maldigestion or malabsorption, the oligomeric formulas with low fat content and high proportion of medium chain fatty acids can be used [12,17]. The rationale of this recommendation was based in experience and in physiological knowledge, but recently, a preliminary randomized clinical trial showed that in critical patients, including those after surgery, the use of peptide-based formulas was associated with fewer gastrointestinal events when they were compared with standard polymeric formulas [18]. 

Moreover, the surveyed surgeons accepted the use of oligomeric diets after gastrectomy or pancreaticoduodenectomy when maldigestion is suspected. Although there are no specific recommendations in the ESPEN guidelines devoted to surgery [14] or pancreaticoduodenectomy [19], it is a common practice to prescribe oligomeric formulas in this situation, even if the oral route is chosen and the usual diet is not sufficient to reach the protein and energy demands of the patient. 

The surgeons also agreed to consider oligomeric formulas in low-debit ileus or colon fistulas. There is no expert consensus regarding the use of semi-elemental formulas when EN is chosen as the medical nutrition therapy in these situations, but data reported in some clinical cases show that this election is a usual practice in enterocutaneous fistula [20,21]. Moreover, interest is growing related to the delivery of nutrients to the distal intestinal limb when a high output proximal enterocutaneous fistula develops. This nutrition strategy, named fistuloclysis, takes advantage of the distal intestine for the infusion of nutrients and chyle drained by the upper intestine. Clinical guidelines, such as the American Society for Parenteral and Enteral Nutrition (ASPEN), recommend using polymeric formulas first and to change to oligomeric formulas if there is intolerance [22]. As they are partially digested, it is supposed that there is a higher nutrient absorption in this intestinal tract. This recommendation is based on case series reporting the advantage of fistuloclysis and EN with the election of semi-elemental formulas in patients suffering from this complication [23,24]. 

There was agreement in the good tolerance of oligomeric formulas in right colectomy or total colectomy. There is a paucity of published reports dealing with EN and semi-elemental formulas in this clinical condition. Diarrhea, due to a fast bowel transit time, is one of the most common complication in the early phase after total of right colectomies and affects up to 46% of patients [25,26]. Usual treatments include dietary changes, antiperistalsis drugs and cholestyramine in the right colectomy with persistence of the distal colon. Therefore, in clinical practice, oligomeric formulas are used in malnourished patients when diarrhea persists, as it has been stated in this survey, and in the experience of the surveyed surgeons, they are well tolerated. We only found one study comparing polymeric formulas with or without fibers after laparoscopic colectomy [27], and more patients had formed stool in the group assigned to the formula with fiber. Thus, more research is needed in this area. Two studies have evaluated the role of elemental formulas (containing complete digested proteins) in the recovery after colectomy and in colorectal surgery complicated with anastomotic leakage. Both studies had, as comparator, treatment with parenteral nutrition. The first one reported a diminished estimated minimum length of stay (LOS) for the recovery associated to the prescription of the elemental formula [13], and in the second situation, the duration of parenteral nutrition was shorter when the elemental formula was used after this complication [28]. To our knowledge, there are no studies evaluating semi-elemental formulas under these circumstances.

The surgeons did not agree on the placement of a feeding tube during the surgical act, usually a needle catheter jejunostomy distal to the anastomosis. ESPEN guidelines devoted to clinical nutrition in surgery advise to consider this strategy in malnourished patients submitted to major upper gastrointestinal surgery or pancreatectomy [14], but in those regarding Enhanced Recovery After Surgery (ERAS) in pancreaticoduodenectomy, the recommendation is to feed with a tube only when an early oral diet has failed, and do not consider systematically to place a jejunostomy [19]. Indeed, controversy exists among surgeons related to this topic. Different studies have shown the feasibility and advantages of EN using a feeding tube distal to the anastomosis, with the tip passed distally in the surgical act. However, there is no consensus about the routine use of a needle catheter jejunostomy, and some authors consider it only for high-risk patients [29]. A recent paper described that more than 8000 patients that had undergone esophagectomy presented with lower LOS, in-hospital mortality and 30-day mortality in the 45% concurrent surgical jejunostomies, compared with no tube placement during surgery [30]. On the other hand, one study using propensity score matching (139 per group) reported a higher incidence of bowel obstruction in patients with jejunostomy tube feeding, while there was no difference in the nutritional status between groups [31]. After gastrectomy, more incidence of complications has been related to jejunostomy, calling individualization to attention [29,32], and in pancreatic surgery, the routine use of distal EN has been abandoned [33,34]. Thus, this is an area of uncertainty, where individualization and the type of intervention required (gastrectomy, esophagectomy or pancreatectomy) may lead to different outcomes after the placement of a jejunostomy. However, the surgeons agreed on the usefulness of early jejunal EN with oligomeric formula in patients undergoing upper gastrointestinal surgery. This is of importance in undernourished patients not meeting oral requirements after surgery [14]. 

The election of oligomeric formulas when EN is provided distally to the anastomosis after upper gastrointestinal surgery is currently practiced among the surveyed surgeons, even though the ESPEN guidelines about surgery state that it is possible to feed the patients with a polymeric standard formula in this situation [14]. There is a lack of evidence about this topic, and in addition to the physiological mechanisms regarding absorption of nutrients when duodenum has been bypassed, novel potential advantages have been published. Thereby, the infusion of EN through jejunostomy using elemental diets with low fat content and MCT has been associated to a non-significant reduction in the incidence of chyle leak after esophagectomy [35]. Some authors advocate the use of low-fat formulas with MCT in jejunal EN after pancreatic surgery in patients at high risk of developing chyle leak and call for more research on this topic [36]. 

The panelists also reached a positive consensus regarding the effectiveness of early jejunal EN compared to intravenous fluid therapy. This agreement is in line with ESPEN clinical guidelines in surgical patients, which state that, whenever possible, oral or EN should be preferred [14]. Scientific evidence from the medical literature shows that an oligo-elemental diet is non-inferior to parenteral or amino acid-based diets regarding digestion, nutrient absorption and tolerance in nutritionally risk patients with Crohn’s disease, SBS and acute and chronic pancreatitis [37].

There are some limitations in this study. Participants were from Spain and represented views and clinical practices within the country. It would be relevant to explore opinions from international expert participants regarding EN in surgical patients with intestinal diseases. Another limitation was the lack of clinical protocols about SBS in most centers, which makes it difficult to homogenize the clinical practice in different hospitals. Finally, a bigger sample of panelists would have been desirable. However, the multicenter consensus reached is potentially applicable for helping the decision-making process in the clinical practice.

## 5. Conclusions

Our study reports a practical approach to the use of oligomeric enteral formulas in surgical patients with intestinal disorders and impaired nutrient digestion or absorption. Our results could help clinicians and surgeons in the decision-making process, to reduce heterogeneity in clinical practice. Further studies and randomized clinical trials are needed to evaluate the use of oligomeric formulas in surgical patients with intestinal diseases.

## Figures and Tables

**Table 1 nutrients-13-01922-t001:** Characteristics of the Delphi panelists and their healthcare facilities.

General Participant Characteristics		N (%)
Gender	Females	11 (23.4%)
	Males	36 (76.6%)
Surgery Unit	Metabolic and Bariatric	1 (2.1%)
	Colorectal	3 (6.4%)
	Esophagogastric	5 (10.6%)
	General and Digestive	31 (66%)
	General and Endocrine	1 (2.1%)
	Hepatobiliary and Pancreatic	6 (12.8%)
Clinical Position	Assistant	30 (63.8%)
	Section Chief	13 (27.7%)
	Head of Service	4 (8.5%)
Number of hospital beds	<100	1 (2.2%)
	100–250	11 (23.9%)
	250–500	15 (32.6%)
	500–1000	17 (37%)
	>1000	2 (4.3%)
Nutrition Unit	No	14 (29.8%)
	Yes	33 (70.2%)
Nutritional protocols for SBS	No	34 (72.3%)
	Yes	13 (27.7%)

SBS: Short bowel syndrome.

**Table 2 nutrients-13-01922-t002:** Statements with the descriptive statistics and the consensus among the participants.

	Statement	Frequency	Median	IQR	Consensus (YES/NO)
Question 1A	Can you assess your knowledge about the article “*Pironi L et al. ESPEN endorsed recommendations. Definition and classification of intestinal failure in adults. Clin Nutr.* 2015; 34(2):171–80” [9].	47	3	3	YES
Question 1B	Can you assess the degree of application in your clinical practice of this article “*Pironi L et al. ESPEN endorsed recommendations. Definition and classification of intestinal failure in adults. Clin Nutr.* 2015; 34(2):171-80” [9].	46	3	3	YES
Question 2	Rate the degree of application in your clinical practice of the SBS protocol used in your center (if applicable).	46	3	3	YES
Question 3	ESPEN clinical guidelines state that a small amount of peptide-based EN is an option to start enteral or oral nutrition in the post-operative hyper-secretion phase of SBS in patients that cannot tolerate polymeric formulas [6].	46	8	1	YES
Question 4	Oligomeric low-fat formulas are effective for patients with nonspecific diarrhea [12].	46	7	2	YES
Question 5	In patients undergoing gastrectomy, especially with a Roux-en-Y anastomosis, oligomeric formula can be beneficial when maldigestion is suspected.	46	7	2	YES
Question 6	Oligomeric formulas can be beneficial in patients treated with cephalic pancreaticoduodenectomy with maldigestion.	46	8	1	YES
Question 7	Oligomeric formulas can be beneficial in patients with low-debit ileus or colon fistulas.	46	8	2	YES
Question 8	Oligomeric formulas are well tolerated in patients treated with right hemicolectomy or total colectomy [13].	46	8	2	YES
Question 9	ESPEN guidelines recommend the use of needle catheter jejunostomy (NCJ) in patients undergoing major abdominal surgery [14].	46	5	5	NO *
Question 10	In clinical practice, it is common to place a feeding tube distally to the anastomosis in the surgical act.	46	4	5	NO *
Question 11	In patients with suspected malabsorption/maldigestion, after upper digestive tract surgery, oligomeric formulas may be useful when EN is carried out distally to the duodenum.	46	8	1	YES
Question 12	Early jejunal EN with oligomeric formulas provide more nutrients with less weight loss compared to intravenous fluid therapy.	46	9	1	YES
Question 13	In patients with suspected malabsorption/maldigestion, early jejunal EN with oligomeric formulas is useful in patients undergoing major upper gastro-intestinal tract surgery.	46	8	2	YES

* This question was asked again in a dichotomous way (yes/no) to reach an agreement in the second round. EN: enteral nutrition; SBS: Short Bowel Syndrome IQR: interquartile range.

**Table 3 nutrients-13-01922-t003:** Statements with descriptive statistics according to the existence of a Nutrition Unit in the Hospital.

	Hospitals with Nutrition Unit			Hospitals without Nutrition Unit		
	Frequency	Median	IQR	Frequency	Median	IQR
Question 1A	14	3	6	32	3	6
Question 1B	14	3	5	32	3	4
Question 2	14	1	1	32	3	6
Question 3	14	8	2	32	7	1
Question 4	14	7	2	32	7	1
Question 5	14	7	2	32	8	2
Question 6	14	7	1	32	8	2
Question 7	14	8	2	32	8	2
Question 8	14	8	2	32	8	2
Question 9	14	4	5	32	5	4
Question 10	14	3	5	32	5	5
Question 11	14	8	2	32	8	2
Question 12	14	8	1	32	9	1
Question 13	14	8	1	32	8	2

IQR; Interquartile range.

**Table 4 nutrients-13-01922-t004:** Descriptive statistics according to the existence of a clinical protocol to approach the diagnosis and intervention of patients with SBS.

Hospitals Without Nutritional Support Protocols in SBS				Hospitals with Nutritional Support Protocols in SBS		
	Frequency	Median	IQR	Frequency	Median	IQR
Question 1A	33	3	6	13	4	5
Question 1B	33	3	5	13	4	3
Question 2	33	1	1	13	7	3
Question 3	33	8	2	13	7	1
Question 4	33	7	2	13	7	1
Question 5	33	7	2	13	7	2
Question 6	33	8	1	13	7	2
Question 7	33	8	2	13	8	1
Question 8	33	8	2	13	8	1
Question 9	33	5	5	13	4	4
Question 10	33	4	5	13	2	4
Question 11	33	8	2	13	8	1
Question 12	33	9	1	13	8	1
Question 13	33	8	1	13	8	2

IQR; Interquartile range.

## Data Availability

The data presented in this study are available on request from the corresponding author. The data are not publicly available due to ethical reasons.

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
