# Peer review of "Oligomeric Formulas in Surgery: A Delphi and Consensus Study"

_nutrients, 2021, doi:10.3390/nu13061922_

Round 1
Reviewer 1 Report
Indication for oligomeric diets in clinical practice is a controversial issue from years. Authors perform a new methodologic approach to this field. Delphi rounds results indicate some interesting points that can serve as a basis for better use of enteral nutrition in surgical patients with intestinal failure.
The work is methodologically well performed. Results are clear. Limitations are also well indicated in the manuscript.
Author Response
Thank you very much for reviewing the manuscript. Spelling has been checked and minor changes have been made.Reviewer 2 Report
This is a very well designed and performed study, clearly presented and described of high clinical importance.
In the introduction - I would suggest to explain clearly the difference between intestinal insufficiency and intestinal failure.
Author Response
Thank you very much for reviewing the manuscript.
The definitions of intestinal failure and intestinal insufficiency have been added to the introduction section:
Intestinal failure (IF) - the reduction of gut function below the minimum necessary for the absorption of macronutrients and/or water and electrolytes, such that intravenous supplementation is required to maintain health and/or growth- and intestinal insufficiency (II) - the reduction of gut absorptive function that doesn't require intravenous supplementation to maintain health and/or growth- [2]